# Older people's challenges and expectations of healthcare in Ghana: A qualitative study

**Confidence Alorse Atakro**[1]\*, **Abigail Atakro**[2], **Janet Sintim Aboagye**[1], **Alice Aluwah Blay**[1], **Stella Boatemaa Addo**[1], **Dorcas Frempomaa Agyare**[3], **Peter Adatara**[4], **Kwaku Gyimah Amoa-Gyarteng**[5], **Awube Menlah**[6], **Isabella Garti**[6], **George Sedinam Boni**[7], **Osei Kwaku Berchie**[1], **Isaac Kwadwo Ansong**[6]

**1** School of Nursing and Midwifery, Christian Service University College, Kumasi, Ghana, **2** Tema General Hospital, Accra, Ghana, **3** University of Cape Coast, Cape Coast, Ghana, **4** School of Nursing and Midwifery, University of Health and Allied Sciences, Ho, Ghana, **5** Suntreso Government Hospital, Kumasi, Ghana, **6** Valley View University, Accra, Ghana, **7** Volta Regional Hospital, Ho, Ghana

\* confidenceatakro@gmail.com

**Data Availability Statement:** Due to the sensitive nature of data, the institutional review board recommended that the data should only be shared on request. Data contain potentially identifying and

## Abstract

### Background

The increase in the number of elderly persons in developing countries has not had a corresponding increase in social and health care support systems for the elderly. There is a substantial difference in the quality of healthcare received by older people in developing and developed countries. Elderly persons in developing countries including Ghana are increasingly becoming marginalised and isolated. There is, however, limited evidence of healthcare challenges and expectations by elderly persons in Ghana. This study explored healthcare challenges and expectations of elderly persons to inform policy that could lead to improved quality of life for elderly persons in Ghana.

### Materials and methods

Qualitative exploratory descriptive study design was used in conducting this study. Semi-structured interviews were used in collecting data from 30 participants from three regions in Ghana (10 from each region). Data analysis was carried out through content analysis.

### Results

Four themes were extracted from data. These themes were: 1. Inadequate information from health workers regarding care of the older person. 2. Queuing frustrations. 3. Financial burden. 4. Focused elderly care demand.

### Conclusion

The elderly in Ghana experience challenges of healthcare which include inadequate information, queuing frustrations and financial burdens. Elderly persons also have expectations of healthcare which include having dedicated professionals and units that will attend to them during their hospital visits. Academic and clinical gerontology experts could collaborate and help improve gerontology knowledge of health workers through workshops and

sensitive patient information and as such the ethical review board had requested that the data only be shared with justifiable reasons. These restrictions were imposed by the institutional review board of the Kwame Nkrumah University of Science and Technology. Data request may be sent through this phone number and email of the ethical review board administrator +233205453785, chrpe@Knust.edu.gh. However, a minimal data set and an audit trail has been provided to enable replication.

**Funding:** The authors received no funding for this work. All funds for the study was provided by the first author.

**Competing interests:** The authors have declared that no competing interests exists.

conferences. Improving knowledge of health workers in gerontology may be a positive step towards meeting healthcare expectations of older Ghanaians.

## Background

The United Nations (UN) defines the elderly as a persons who is 60 years or above [1]. The world's population of elderly persons will nearly double from 12% to 22% between 2015 and 2050 [2] with 80% expected to reside in low and middle-income countries (LMICs) by 2050 [1]. Elderly persons globally suffer from various health problems such as chronic conditions, injuries, depression from loneliness, malnutrition, visual problems, hearing loss and complex dental problems [3–5]. Elderly persons across the globe are increasingly becoming isolated and marginalized despite the several challenges they face [6]. These challenges have been found to be proportionately higher in developing and LMICs due to social and economic disadvantages [7, 8]. The Challenges of elderly people in developing and LMICs are due to wide global variations of inequalities related to quality of life of elderly persons [9]. Whilst developed countries are increasingly providing opportunities for quality healthcare of their older population through services such as residential and communities services [10], developing countries lack such services for their elderly people [11]. In contrast to developed nations, several parts of the developing world have the majority of their elderly persons in rural areas and urban slums with no pension schemes [12, 13]. The response of African governments to issues of economic, health, disability, and living conditions in old age are minimal when compared with governments' response on other continents [14–16]. The recommended societal approach to population ageing, which includes building an age-friendly health system, requires a transformation of health systems in sub-Saharan Africa from curative models to provision of integrated care that is centered on preventive, psychosocial and cultural needs of elderly persons [2, 17]. There is, however, limited research evidence on long-term trends in psychological, cultural, health and social support systems of elderly persons in Sub-Saharan African countries [18]. The developmental contributions that can be derived from elderly persons in African countries will heavily depend on the quality of healthcare provided to them.

The proportion of elderly persons in Ghana is currently at 7.2 percent, which indicate that Ghana has one of the highest proportions of elderly persons in sub-Saharan Africa [15]. The increase in the number of elderly persons has not had a corresponding increase in social and health care support systems for the elderly [15]. Population ageing is occurring at a time in which traditional systems that supported elderly persons have been affected by modernisation and globalisation resulting in downward trends of support through public welfare systems [16]. Existing inequalities in elderly care is particularly true for Ghana as two-third of her elderly population live in rural settings and are vulnerable to greater socio-economic and health marginalisation [19]. The government of Ghana tends to invest in health, education, and social support systems for her younger population whilst neglecting the support systems needed for elderly persons who contribute immensely to the development of their communities, families and country [18, 20].

Although the National Health Insurance Scheme (NHIS) of Ghana by law covers elderly persons above 70 years [21], elderly persons above 70 years are faced with the challenge of bearing the cost of health services, for example, paying for medications and investigations due to the inability of the National Health Insurance Authority (NHIA) to remit health facilities on time. Elderly persons within the age brackets of 60 and 69 have no NHIS cover, although the compulsory retirement age in Ghana is 60 years and average life expectancy is

63 years [14, 21]. Ghana may miss the opportunity of deriving maximum developmental benefits from her elderly population if she fails to make the appropriate adaptations and investments in the elderly [18]. The role of healthcare professionals in improving health outcomes of elderly persons in Ghana cannot be underestimated [22]. However, several studies indicate that healthcare workers in Ghana lack the necessary knowledge in the care of the elderly [11, 13, 23, 24].

Inadequate knowledge in gerontology by healthcare workers usually results in negative health outcomes for elderly persons' [23, 25]. Ghana currently lacks age friendly hospitals and gerontological experts [26]. Very little is known about the health of the elderly population in Ghana. Although district hospitals are the first point of contact for ill elderly persons in Ghana, research evidence on the challenges and expectations of elderly persons in these hospitals is seldom looked into. Several reports on older people's care in Africa conclude there is an urgent need to constitute an agenda of research on ageing in African countries [16, 27–29].

This study set out to inform Ghanaian elderly care policy by identifying:

1. Challenges of elderly persons regarding healthcare in the Ghanaian health system, and

2. Expectation of elderly persons regarding healthcare in the Ghanaian health system.

A qualitative approach was used in this study because authors wanted to understand details older people's challenges and expectations of healthcare, from the viewpoint of older people themselves.

## Materials and methods

### Study design

Qualitative exploratory descriptive design was used in conducting this study. Qualitative exploratory descriptive design was found to be useful in exploring experiences of older people in healthcare [30]. The design enabled researchers in this study to have in-depth understanding into challenges and expectations of healthcare by elderly persons in Ghana. The study was reported using the COREQ criteria for reporting qualitative research [31].

### Study setting

Ghana is a country in West-Africa with a current projected population of 30.3 million people [32]. Ghana is divided into 16 regions. These 16 regions are divided into northern, middle and southern zones. The Northern zone consist of Upper West, Upper East, Northern, Northern East, and Savannah Regions [33]. The middle zone consist of Brong Ahafo, Bono East, Ahafo and Ashanti Regions [33]. The southern zone is made up of Western, Western North, Central, Greater Accra, Volta, Oti, and Eastern Regions [33]. This study was undertaken in three purposely selected regions in Ghana: one region from the southern zone, one from the middle zone and the other from the northern zone. Volta Region was selected from the Southern Zone, Ashanti from the middle zone and the Upper West from the Northern Zone. There are 17 district hospitals in the Volta Region, 25 district hospitals in the Ashanti Region, and 3 district hospitals in the upper west Region [33]. The bed capacities of public health hospitals in the Ashanti, Upper West and Volta are 1230, 225, and 1400 respectively [33]. Data was collected from these three regions because researchers wanted to have an indepth understanding into challenges and expectation of elderly persons in many areas of Ghana.

## Population and sampling

A purposive sampling technique was used in selecting one region from each of the three zones in Ghana. A purposive sampling technique was utilised in selecting regions because the researchers wanted to include a region from each of the three regional zones (northern zone, middle zone and southern zone) in Ghana. The Ashanti, Volta and Upper West Regions were selected from each zone. Participants in these regions were also selected through a purposive sampling technique. The targeted population for this research were persons who were 60 years and above, who had visited district hospitals in selected regions. These persons were selected in line with the United Nations (UN) definition an elderly person. The UN defines the elderly as a persons who is 60 years or above [1]. All participants were within the ages of 60 and 89 years. Only elderly persons who had visited the hospital during the year were involved in study. Participants were recruited through district focal persons of the Ministry of Gender and Social protection. Letters were sent to older persons requesting their participation in study. Only older persons who indicated their willingness to participate and met inclusion criteria were contacted for data collection. Only persons who could speak English were included in study. Older people who had intellectual disabilities or had mental illnesses were excluded from study.

## Data collection

Data were collected with a semi-structured interview guide which was formulated by the research team (S1 Appendix). The semi-structured interviews were conducted in English since all participants could speak English. The interview guide was pretested on ten older people in other regions to identify ambiguous questions. Questions asked during interviews included the following: 1. Can you describe any challenges from healthcare workers that you faced when you visited a district hospital here for treatment? 2. Describe your perception of healthcare in Ghanaian district hospitals. 3. Describe how you see the current care of the elderly in Ghanaian hospitals. 4. Describe ways you think health workers can improve their care for you in the hospital. Probes were used to elicit further descriptions of challenges and expectations. Data were collected within a three-month period from December 2018 to February 2019. Each interview lasted between one and two hours. Five participants were initially interviewed in each region. Additional five persons were interviewed in each region as saturation was not determined with the initial interviews. Saturation was determined after interviewing the 30th participant. The number of participants selected for qualitative interviews depends on the purpose of the study [34]. Qualitative studies work with small numbers that are feasible to study in depth [34]. Transcribed data was shown to some participants for their comments. Transcribed data were securely stored on a flash drive which was password protected.

## Data analysis

Data analysis was conducted with qualitative content analysis. Data was analysed manually by research team which was headed by a professor of nursing. Data was analysed in line with COREQ criteria for analysing qualitative research data [31]. The COREQ criterion for analysing qualitative data uses the following pattern: coding; derivation of categories from codes; formation of themes; participants checking of codes; presentation of quotations. The research team sat together and read through the content of field data. Data was cleaned by removing all identifiable information. Codes were found during readings of transcripts. Codes were discussed within the research team. Similar codes were used by the team to create families and similar families grouped together as themes. The themes were discussed among all members

of the research team for agreement. The themes were also discussed with some participants to find out if themes represented their views.

## Rigour

A pretest of semi-structured interview guide was carried out in the Central Region. Pretest ensured that ambiguous questions were modified to make them clearer for participants. The researchers had prolonged interactions with elderly persons to ensure in-depth understanding of findings that emerged. Member checking was carried out to validate data from the participants. Data transcriptions and coding were done by the research team to ensure that the right challenges and expectations were reported. Researchers went back to elderly persons to find out if themes formulated represented their opinions. The background of the authors as qualitative researchers further helped in providing qualitative research rigour in the study.

## Ethical consideration

Ethical approval for this study was granted by the Committee on Human Research Publication and Ethics (CHRPE) at the Kwame Nkrumah university of Science and Technology (KNUST), Ghana with reference number CHRPE/AP/634/18. Permissions were also sought from district assemblies where data were collected. Anonymity and confidentiality were explained to participants. Participants were assured that withdrawal from study will not in any way attract sanctions. Informed consent forms were filled and signed by participants. Participants were identified with codes to ensure anonymity. Questions that could cause any form of psychological trauma on participants were avoided.

## Results

### Demographic data of respondents

As shown in Table 1, the majority of participants were women. All participants were within the ages of 60 and 89. The majority of elderly persons were Christians and as many as 40% were widows/widowers. The highest level of formal education for most participants was primary school level (see Table 1). About 67% were retired from their formal jobs.

### Themes

Four themes were formulated from content analysis. The themes were: 1. Inadequate information from health workers regarding care of the older person. 2. Queuing frustrations. 3. Financial burden. 5. Focused elderly care demand.

### Inadequate information from health workers regarding care of the older person

Majority of participants pointed out that inadequate information from health workers in the out-patient department and other departments in Ghanaian district hospitals was a challenge for them. Elderly persons did not have detailed explanations of nursing activities when they visited district hospitals for care. Elderly persons indicated they did not receive specific education from nurses and medical officers regarding maintenance of good health in old age. Some participants attributed the lack of information from nurses and medical officers to inadequate time and workload:

**Table 1. Demographic characteristics of participants.**

| Parameters | Value |
|---|---|
| Age (years) | Number (%) |
| 60–69 | 15 (50%) |
| 70–79 | 10 (33.3%) |
| 80–89 | 5 (16.7) |
| Gender | |
| Male | 12 (40%) |
| Female | 18 (60%) |
| Religion | |
| Christianity | 20 (66.7%) |
| Islam | 7 (23.3%) |
| Traditional religion | 3 (10%) |
| Education | |
| Primary school level | 20 (66.7%) |
| Junior high school level | 6 (20%) |
| Secondary school level | 2 (6.7%) |
| Tertiary level | 2 (6.7%) |
| Marital status | |
| Married | 10 (33.3%) |
| Divorced | 8 (26.7%) |
| Widow/widower | 12 (40%) |
| Employment | |
| Currently employed | 10 (33.3%) |
| Retired | 20 (66.7%) |

*. . .Nurses and doctors did not explain issues into details when taking care of me. Maybe it is because they don't have much time. Not much explanation is given on the needs and how important that task will affect my health. We also need more information on how to prevent diseases from getting to us in our old age*

*[AP3].*

Some participants shared their thoughts on the need for receptions in district hospitals similar to what can be found in several other organisations in Ghana. Others said they wished they could be given necessary information at these proposed receptions without having to ask for such information from nurses and medical officers:

*. . . I think nurses and doctors could give us enough information without asking. If they can have receptions to give us information in the hospital, it will be good*

*[AP5].*

*. . . . . .Some nurses should be at receptions to explain things to us to understand well. Information is necessary to know what to do.. ..we don't have to ask for information all the time before we are offered important information about our health. . .*

*[AP7]*

## Queuing frustrations

The majority of participants stated their frustrations in joining long queues for treatment in Ghanaian district hospitals. Participants shared their thoughts on the need for prioritisation of their healthcare needs because of their age. Participants did not want to join the regular long queues found in many Ghanaian district hospitals:

*I am always joining long queues with the young and energetic people. Health workers keep us in a long queue and keep us waiting before later seeing the doctor. This is frustrating. . ...they should consider treating us with special care*

[AP28]

*. . ...My son look at me and my age and joining all those long queues in the hospital for treatment. If I had an option to find some other treatment somewhere else, I will take it. . .. . .*

[AP3].

The long waiting hours in queues have resulted in some participants seeking to treat themselves with over the counter medications instead of reporting to the hospitals for treatment:

*. . ..It is not easy when you think of going to the hospital. All that comes to mind is the long queues that will be waiting for you when you get there. Sometimes I just go to the drug store or the local chemist shop to get something for myself instead. . .*

[AP3].

Participants indicated that they usually try other sources of treatment locally because of the frustrations they experience in hospitals. However, they visit hospitals when there are complications in their conditions:

*. . .. I prefer to take some drugs at home rather than to waste my time at the hospital. But when it becomes very serious, I go to the hospital*

[AP20].

*. . .Sometimes I just buy some local medicines and take to help myself rather than going to the hospital to queue. But when the local medicines are not helping, I go to the hospital*

[AP1].

## Financial burden

Participants in the study reported that there were high financial burdens in seeking medical treatment in Ghanaian district hospitals. Although Ghana has an insurance system for the elderly persons, participants interviewed stated they had to buy most of the expensive medications that were prescribed in the hospitals and pay for some investigations. Participants indicated the need for subsidisation of their healthcare cost by government:

*Although we have insurance, we still pay for services when we go to the hospital. Many of the good medicines are never on the insurance. Even some of the investigations that we are*

*supposed to do in the hospital are not free. The insurance should cover everything. There should also be a reduction in the cost involved in our care since we don't work at this age. . .*

*[AP4].*

*They say we have insurance covering elderly persons in Ghana but when I go to the hospital, they tell me that I have to still pay for some stuff such as investigations and medicines. The insurance seems to be for only folder. I don't think this should be the case*

*[AP2].*

Participants were financially constrained as a result of retirements from their formal jobs:

*. . ...The government has to improve the insurance especially for elderly persons since our income has reduced due to retirement. I think we should be attended to without all the cost that we bear in district hospitals. The Government can make this happen.*

*[AP 7].*

Some participants who were between the ages of 60 and 70 indicated their frustration at their exclusion from the health insurance package for the aged in Ghana.

*. . .The health insurance is supposed to cater for elderly persons over 70 years. What about those of us who are 65. We retire at 60 in Ghana, but the insurance starts from 70. This I don't understand. . ..*

*[AP29].*

Some participants were of the view that the National Health Insurance Authority in Ghana does not reimburse hospitals regularly and this was partly the cause of having to pay for health-care cost despite being covered by the national health insurance.

*. . .The health insurance people don't pay the hospitals, so hospitals don't also give us the free service they are supposed to provide for us. . .. . ...the hospitals make us pay for services because they don't have any other income*

*[AP9].*

## Focused elderly care demand

The theme of focused elderly care demand has two sub-themes: 1. Need for routine check-up visits. 2. A call for dedicated elderly care units and consulting rooms.

### Need for routine check-up visits

Participants were of the view that problems they faced could be reduced to the barest mini-mum through regular check-up visits from healthcare workers. Participants indicated that these check-up visits could take place in their communities or homes. Check-up visits in their homes or communities will help reduce the cost of transportation to hospitals for healthcare:

*. . . .For me, there should be regular check-up programs for us in our homes. These check-up visits could be free so that many of our challenges can be prevented. It will prevent our travels and monies spent on travelling all the way to the hospitals*

[AP24].

*. . . . . . .I think sometimes these check-up visits can be done in our communities. Sometimes nurses and other health workers can come and see how we are doing in our homes. It will be good for us, so we do not take cars and spend our little monies going to the hospital*

[AP26].

Some participants also recommended that in the absence of the regular visits by health workers, health workers can alternatively call them on the phone on regular basis to check up on them:

*. . .Even if health workers can't come to us regularly, they should call us on the phone on some regular basis to find out how we are doing. We have phones now, so we are ready for health workers to call us and check up on us*

[AP7].

*. . .Check-up visits can even be done through calls to us by health workers, so we know whether we really need to come to the hospital. I am reluctant to take a car all the way to the hospitals for check-up. . . . . . . .*

[AP28].

Participants shared their thoughts on the importance of regular check-up visits from nurses as pertains to monthly antenatal programs for pregnant women in Ghanaian hospitals.

*. . . . . .Just like what is done for pregnant women, we can also have dedicated days to come for our check-up or reviews. Nurses have done it for pregnant women so why can't they do it for us too*?

[AP18].

## A call for dedicated units and consulting rooms

Participants suggested that Ghanaian district hospitals should be designed to prevent queuing for elderly persons. Creation of elderly wards and consulting rooms with aged care experts were suggested by participants. Participants expected health workers to dedicate a consulting room to them to reduce their waiting time at the out-patient department:

*. . . . . .I mean there should be departments and special health workers for elderly persons. The health workers can find a separate room or unit for us where they will see only elderly persons.. . .*

[AP11].

*For us not to queue for longer periods, health workers who have knowledge in old people care should be employed to care for elderly persons. In that case, we will just go straight to see our health workers when we go to the hospital. . .*

[AP9].

Participants were of the view that hospital mangers and health workers could help in setting up these special units and consulting rooms for the elderly client:

*. . .. Managers and health workers of the district hospitals in this country should come together and develop a plan that will make sure that there are units with experts for the elderly in every hospital just for the elderly. . . . . .*

[AP3].

*. . .Our hospitals should create units and consulting rooms just for us to come and see special-ist health workers. . . . . .*

[AP24].

## Discussion

This section is discussed in line with findings of the study. The discussion is therefore orga-nised under the following sub-headings: Inadequate information from health workers regard-ing care of the older person; Financial burden of the older Ghanaian regarding healthcare; Dedicated units and consulting rooms for the care of older Ghanaians.

### Inadequate information from health workers regarding care of the older person

Participants indicated that healthcare workers did not provide them with enough information on how to prevent diseases in their old age. This may be due to inadequate education of health workers in gerontological care in Ghana [11, 29]. Clinical staff such as nurses and medical offi-cers are not examined in gerontological care during their licensing examinations [11, 35]. A policy on inclusion of gerontology content in the training of healthcare workers in Ghana may contribute to addressing issues of inadequate information from health professionals regarding aged care. Healthcare workers in Ghana should be provided knowledge on the current national aging policy of Ghana [36] in order to understand their roles regarding the care of the older Ghanaian. Training could be conducted through workshops to improve knowledge of health-care workers. These ideas are supported by previous studies from other parts of the world [37–40]. These studies [37–40] recommend curriculum modifications and training in gerontology as necessary steps for meeting expectations of elderly persons and achieving positive health outcomes for the elderly.

### Financial burden of the older Ghanaian regarding healthcare

Participants in this study indicated that meeting the current healthcare cost in hospitals is a challenge to them. Currently the National Health Insurance Scheme (NHIS) only covers elderly persons above 70 years [21] whilst life expectancy of the Ghanaian elderly person is 63 years and compulsory retirement is 60 years [14]. The national health insurance scheme could be extended to elderly persons between 60 years and 69 years since many of them may not live to 70 years to benefit from the current elderly health insurance package. Today's generation owe it as a duty to honour and guarantee better living conditions for our elderly persons [21] as they have contributed their quota to the development of the nation [36]. Ageing could also be defined in terms of functionality rather than a stage in a life time because some elderly per-sons could function at the age of 60 and beyond [41, 42].

The fact that 33% of elderly participants in this study were still actively engaged in various forms of employment (Table 1) shows that some older people could work after age 60. Rehiring older people is in line with the United Nation's (UN) recommendation on rehiring retired persons to enhance knowledge transfer to the younger generation of workers [43]. Retiring every elderly person at the age of 60 may be disadvantageous to institutions which could benefit from their experiences. The UN recommends development of regulations for the utilisation of the experiences of elderly persons who are over 60 years [43]. Older people who are active and still working after age 60 may find it easier to pay for their healthcare cost and avoid unorthodox methods of treatments which complicates their conditions. Being active at an old age has also been shown to improve physical and psychological well-being of older people [44–46].

## Dedicated units and consulting rooms for the care of older Ghanaians

In this study, older Ghanaians indicated that they expected hospitals to implement dedicated services towards older people's care. Participants indicated the need for dedicated units and consulting rooms with gerontology experts where they could be provided services. Evidence available shows such dedicated services can reduce complications and admission rates in older people and cost effective for both older people and health facilities [47, 48]. Older persons could have their check-up visits in these dedicated units to avoid queuing with other aged groups. Staff from these dedicated aged care units could provide regular community care to older Ghanaians. Specialist training could be provided to health workers in order to provide such dedicated services to older people. Academic and clinical gerontology experts could help establish advanced specialist courses for health workers in Ghana to be able to manage these dedicated units. Implementing these suggested dedicated units for older people will prevent the frustration of constant queuing, which was a major concern of participants in the study.

Aged care home services could also be introduced by the Ghana Health Service (GHS) as pertains in other countries [49, 50]. The introduction of aged care home services by the GHS should be accompanied by appropriate regulatory and monitoring mechanisms that will protect and improve health of older Ghanaians. This is because increasing commercialisation, lack of regulation and inadequate monitoring can result in poor quality care for elderly persons in aged care homes [50]. The regulatory mechanism could include the use of professionally trained persons and adequate remuneration for these professionals. Elderly care models that are implemented in Ghana should integrate client choices, community/primary health care services, residential and non-residential elderly care services, as these have been found to be effective in other countries [51].

## Strengths of the study

This study discussed challenges and expectation of the elderly regarding healthcare in Ghana. It is one of the few studies that explored this phenomenon in Ghana. The researchers also covered the three zones in Ghana by taking data from a district in the northern, middle, and southern zones. A rigorous result was ensured by including qualitative research experts in this study. This study discussed models of elderly care as pertains to other continents and recommended elderly care models for Ghana. Further education for health workers in gerontology has been suggested in a bid to improve expertise for gerontology care in Ghana.

## Limitations of study

This study investigated only district hospitals which are public hospitals in Ghana. Further studies of private, regional, and tertiary health institutions may also be necessary in the future to compare results to district hospitals. Data from three regions (out of 16) in Ghana may not

be sufficient to make adequate generalisation from study results. It must however be stated that qualitative research is usually assessed regarding credibility, transferability, dependability, and confirmability [52] which were applied and described throughout the study. Transferability to similar population may be possible due to the rigorous qualitative methods used to conduct the study.

## Conclusion

Elderly persons in Ghana have challenges as well as expectations of healthcare. Challenges include inadequate information, queuing, and financial burdens. Expectation of nursing care included organisation of regular routine check-up visits, dedication of units to the elderly and cost subsidisation. Health workers could be trained in gerontological care through workshops to provide appropriate care to older Ghanaians. Bottom up approaches should be utilised in elderly care where concerns and opinions of the elderly and their relatives are considered in their care. Evidence informed curricula frameworks for teaching elderly care in Ghana could be formulated through further research. The current national aged policy could be included in curricula of health training institutions to create some awareness of elderly care in Ghanaian health students.

## Supporting information

**S1 File. Audit trail for aged care research.**
(DOCX)

**S2 File. Interview data.**
(DOCX)

**S1 Appendix. Interview guide.**
(DOCX)

## Acknowledgments

The authors are grateful to older persons who participated in this study.

## Author Contributions

**Conceptualization:** Confidence Alorse Atakro, Abigail Atakro, Janet Sintim Aboagye, Alice Aluwah Blay, Stella Boatemaa Addo, Dorcas Frempomaa Agyare, Peter Adatara, Kwaku Gyimah Amoa-Gyarteng, Awube Menlah, Isabella Garti, George Sedinam Boni, Osei Kwaku Berchie, Isaac Kwadwo Ansong.

**Data curation:** Confidence Alorse Atakro, Abigail Atakro, Janet Sintim Aboagye, Alice Aluwah Blay, Stella Boatemaa Addo, Dorcas Frempomaa Agyare, Peter Adatara, Kwaku Gyimah Amoa-Gyarteng, Awube Menlah, Isabella Garti, George Sedinam Boni, Osei Kwaku Berchie, Isaac Kwadwo Ansong.

**Formal analysis:** Confidence Alorse Atakro, Abigail Atakro, Janet Sintim Aboagye, Peter Adatara, Kwaku Gyimah Amoa-Gyarteng, Awube Menlah, Isabella Garti, George Sedinam Boni, Osei Kwaku Berchie, Isaac Kwadwo Ansong.

**Investigation:** George Sedinam Boni.

**Methodology:** Confidence Alorse Atakro, Abigail Atakro, Janet Sintim Aboagye, Alice Aluwah Blay, Stella Boatemaa Addo, Dorcas Frempomaa Agyare, Peter Adatara, Kwaku Gyimah

Amoa-Gyarteng, Awube Menlah, Isabella Garti, George Sedinam Boni, Osei Kwaku Berchie, Isaac Kwadwo Ansong.

**Writing – original draft:** Confidence Alorse Atakro.

**Writing – review & editing:** Confidence Alorse Atakro, Abigail Atakro, Janet Sintim Aboagye, Alice Aluwah Blay, Stella Boatemaa Addo, Dorcas Frempomaa Agyare, Peter Adatara, Kwaku Gyimah Amoa-Gyarteng, Awube Menlah, Isabella Garti, George Sedinam Boni, Osei Kwaku Berchie, Isaac Kwadwo Ansong.

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
