## [Editor Report · Decision Letter 0]

28 Jan 2020

PONE-D-19-28368

Older people's challenges and expectations of nursing care in Ghana: A qualitative Study.

PLOS ONE

Dear Prof Atakro,

Thank you for submitting your manuscript to PLOS ONE. After careful consideration, we feel that it has merit but does not fully meet PLOS ONE’s publication criteria as it currently stands. Therefore, we invite you to submit a revised version of the manuscript that addresses the points raised during the review process.

See comments below.

We would appreciate receiving your revised manuscript by Mar 13 2020 11:59PM. To enhance the reproducibility of your results, we recommend that if applicable you deposit your laboratory protocols in protocols.io, where a protocol can be assigned its own identifier (DOI) such that it can be cited independently in the future. For instructions see: http://journals.plos.org/plosone/s/submission-guidelines#loc-laboratory-protocols

We look forward to receiving your revised manuscript.

Kind regards,

Andrew Soundy

Academic Editor

PLOS ONE

Journal Requirements:

3. Please include additional information regarding the aemi-structured interview guide used in the study and ensure that you have provided sufficient details that others could replicate the analyses. For instance, if you developed a guide as part of this study and it is not under a copyright more restrictive than CC-BY, please include a copy, in both the original language and English, as Supporting Information. Also, pretesting of this guide was referred to in the text but further details concerning the nature of number of participants is required. Furthermore, please expand on the recruitment of participants to this study and any inclusion/exclusion criteria employed.

Additional Editor Comments:

Abstract

No need for prevalence figures in background information

Introduction

Make sure you justify why qualitative descriptive approaches are so valuable

Methods

You need to consider a framework which will support your write up e.g., COREQ or SRSQ – you need this to make sure you address each requirement of these – again it is not about a correct answer but that your method make sense in terms of the results you give e.g., sample size considerations? when I started to look at this type of approach this article talks about content analysis and frameworks https://journals.sagepub.com/doi/full/10.1177/1937586715614171

Study design

You will need to give a better reference for your qualitative descriptive design e.g., a paper which gives key steps that are expected to be taken e.g., a specific paper which mentions some options could be. A general text book which is not specific to the design wouldn’t be appropriate. Reconsider this and within this consider your paradigmatic stance and methodological stance

You need to justify your selected interview guide – did you do a cognitive interview before? Was it piloted – did you use literature to work out questions? Why so few questions? Not looking for a correct answer but your choices need justification

You will need to include a full audit trail without this it will be hard to follow how you got the results that you have obtained - without this the paper will be rejected at the next point.

Does your data analysis approach fit with the descriptive design you mention above? Please justify your choices

Results

I would like to see themes at the minor level from an audit trail
---

## [Author Response · Author response to Decision Letter 0]

12 Mar 2020

Comment 1:When submitting your revision, we need you to address these additional requirements.

Response 1: Plos one requirements were followed in the submission of manuscript

Comment 2: We note that you have indicated that data from this study are available upon request. PLOS only allows data to be available upon request if there are legal or ethical restrictions on sharing data publicly. For information on unacceptable data access restrictions, please see http://journals.plos.org/plosone/s/data-availability#loc-unacceptable-data-access-restrictions.

Response 2: Cover letter was modified to reflect data availability. An audit trail has now been provided that shows data collection at every stage and could show some challenges and expectations described by older people. Minimal data collected has also been provided to enable replication of this study. The number of the ethical committee provided for request of full data set.

Comment 3:Please include additional information regarding the semi-structured interview guide used in the study and ensure that you have provided sufficient details that others could replicate the analyses. For instance, if you developed a guide as part of this study and it is not under a copyright more restrictive than CC-BY, please include a copy, in both the original language and English, as Supporting Information. Also, pretesting of this guide was referred to in the text but further details concerning the nature of number of participants is required. Furthermore, please expand on the recruitment of participants to this study and any inclusion/exclusion criteria employed.

Response 3: A copy of interview guide provided. all participant could speak English and did not need any translation for researchers. Additional information on the pretest such as the number pf people pre-tested has now been added.

Comment 4:Abstract

No need for prevalence figures in background information

Response 4: The prevalence has now been removed as suggested.

comment 5: Introduction

Make sure you justify why qualitative descriptive approaches are so valuable

Response 5:Justification of the qualitative study in introduction has now been included and highlighted.

Comment 6: . Methods

You need to consider a framework which will support your write up e.g., COREQ or SRSQ – you need this to make sure you address each requirement of these – again it is not about a correct answer but that your method make sense in terms of the results you give e.g., sample size considerations? when I started to look at this type of approach this article talks about content analysis and frameworks https://journals.sagepub.com/doi/full/10.1177/1937586715614171

Response 6: The COREQ framework has now been used in reporting work. The framework of COREQ was used in reporting all sections of the study.

Comment 7: Study design

You will need to give a better reference for your qualitative descriptive design e.g., a paper which gives key steps that are expected to be taken e.g., a specific paper which mentions some options could be. A general text book which is not specific to the design wouldn’t be appropriate. Reconsider this and within this consider your paradigmatic stance and methodological stance

Response 7: This suggestion has been complied with and highlighted.

Comment 8: You need to justify your selected interview guide – did you do a cognitive interview before? Was it piloted – did you use literature to work out questions? Why so few questions? Not looking for a correct answer but your choices need justification.

Response 8: Justification now provided for the interview guide and highlighted. Questions were developed based on literature review. A pilot was done with 10 older people and ambiguous questions were modified.

Comment 9:You will need to include a full audit trail without this it will be hard to follow how you got the results that you have obtained - without this the paper will be rejected at the next point.

Response 9: The audit trail of this study has now been provided. 

Comment 10: Does your data analysis approach fit with the descriptive design you mention above? Please justify your choices.

Response 10: This suggestion has been complied with and highlighted.

Comment 11: . Results

I would like to see themes at the minor level from an audit trail

Response 11: Audit trail now provided.

---

## [Decision Letter · Decision Letter 1]

7 May 2020

PONE-D-19-28368R1

Older people's challenges and expectations of nursing care in Ghana: A qualitative Study.

PLOS ONE

Dear Dr Atakro,

Thank you for submitting your manuscript to PLOS ONE. After careful consideration, we feel that it has merit but does not fully meet PLOS ONE’s publication criteria as it currently stands. Therefore, we invite you to submit a revised version of the manuscript that addresses the points raised during the review process.

Please see comments from the reviewers and address. 

We would appreciate receiving your revised manuscript by 6 June 2020. To enhance the reproducibility of your results, we recommend that if applicable you deposit your laboratory protocols in protocols.io, where a protocol can be assigned its own identifier (DOI) such that it can be cited independently in the future. For instructions see: http://journals.plos.org/plosone/s/submission-guidelines#loc-laboratory-protocols

We look forward to receiving your revised manuscript.

Kind regards,

Andrew Soundy

Academic Editor

PLOS ONE

Reviewers' comments:

Reviewer's Responses to Questions

**Comments to the Author**

1. If the authors have adequately addressed your comments raised in a previous round of review and you feel that this manuscript is now acceptable for publication, you may indicate that here to bypass the “Comments to the Author” section, enter your conflict of interest statement in the “Confidential to Editor” section, and submit your "Accept" recommendation.

Reviewer #1: (No Response)

Reviewer #2: (No Response)

2. Is the manuscript technically sound, and do the data support the conclusions?

Reviewer #1: Yes

Reviewer #2: (No Response)

3. Has the statistical analysis been performed appropriately and rigorously? 

Reviewer #1: N/A

Reviewer #2: (No Response)

4. Have the authors made all data underlying the findings in their manuscript fully available?

Reviewer #1: Yes

Reviewer #2: (No Response)

5. Is the manuscript presented in an intelligible fashion and written in standard English?

Reviewer #1: Yes

Reviewer #2: (No Response)

6. Review Comments to the Author

Reviewer #1: 1. Study setting

Page 5 under settings: lines 2 -5, the authors stated that Ghana has 10 regions and listed the 10 regions. Now Ghana has 16 Regions. I recommend that the 16 regions should be used since the work is yet to be published. When stating the regions, for instance Ashanti Region, the R for the region should be an upper case. For example, Volta Region, Ashanti Region etc.

2. Data collection

Page 6, line 2, I suggest that the authors use researchers instead of investigators.

3. Page 6, line 4: the authors stated that the interview guide was tested on 10 elderly persons; were these 10 elderly persons selected from the 3 research settings or other settings? Please state it.

4. Pages 6 and 7 seem to have repetitions:

• lines 3-6, the authors stated that “The interview guide was developed based on literature and pre-tested on 10 elderly persons in order to identify and modify ambiguous questions. Semi-structured interviews (Appendix 1) were conducted in English since all participants could speak English.”

• Last sentence on page 6 and first sentence on page 7 seems to be saying the same thing; “Interview guide was pretested on ten older people to identify ambiguous questions. Interviews were conducted in English”.

The authors should please work on the repetition.

5. Rigour

Page 7 line 1 should be “a pretest of semi-structured interview guide was …” the guide was omitted.

6. Ethical considerations

Page 8, the last sentence: the authors stated that “Researchers made sure that this study did not cause any physical or psychological harm to any participant”. Please state how these were ensured.

7. Demographic characteristics of participants

Page 8, the authors should be consistent with the use of the percentages. Some are in words and others are in figures. Please reconcile.

8. Discussions

• Page 17: instead of using ‘elderly persons in this study’ authors should rather use participants. This should be effected throughout the manuscript.

• Page 19, line 20: The authors should use client instead of patient since not all aged sent to the aged homes for care are sick.

9. Strengths of the study

Page 20 line 4: throughout the manuscript, I did not see any recommendation for medical officers to be trained in Gerontology as stated here by the authors. Please delete the medical officers if it has not earlier been stated.

10. There are few grammatical errors. I suggest the authors employ the services of a language editor

Reviewer #2: (No Response)

7. PLOS authors have the option to publish the peer review history of their article (what does this mean?). If published, this will include your full peer review and any attached files.

Reviewer #1: No

Reviewer #2: No

---

## [Author Response · Author response to Decision Letter 1]

19 May 2020

Response to reviewer 1

comment 1:Study setting

Page 5 under settings: lines 2 -5, the authors stated that Ghana has 10 regions and listed the 10 regions. Now Ghana has 16 Regions. I recommend that the 16 regions should be used since the work is yet to be published. When stating the regions, for instance Ashanti Region, the R for the region should be an upper case. For example, Volta Region, Ashanti Region etc.

Response 1: These recommended corrections have been adhered to and highlighted in manuscript.

Comment 2: Data collection

Page 6, line 2, I suggest that the authors use researchers instead of investigators.

Response 2: This suggestion had been adhered to and highlighted.

Comment 3: Page 6, line 4: the authors stated that the interview guide was tested on 10 elderly persons; were these 10 elderly persons selected from the 3 research settings or other settings? Please state it.

Response 3:The pretest was conducted in another region (Central region). This is now stated in manuscript as suggested.

Comment 4:Pages 6 and 7 seem to have repetitions:

• lines 3-6, the authors stated that “The interview guide was developed based on literature and pre-tested on 10 elderly persons in order to identify and modify ambiguous questions. Semi-structured interviews (Appendix 1) were conducted in English since all participants could speak English.”

• Last sentence on page 6 and first sentence on page 7 seems to be saying the same thing; “Interview guide was pretested on ten older people to identify ambiguous questions. Interviews were conducted in English”.

The authors should please work on the repetition.

Response 4: Repetitions were removed throughout manuscript.

Comment 5:Rigour

Page 7 line 1 should be “a pretest of semi-structured interview guide was …” the guide was omitted.

Response 5:This recommended correction has been adhered to and highlighted.

Comment 6: Ethical considerations

Page 8, the last sentence: the authors stated that “Researchers made sure that this study did not cause any physical or psychological harm to any participant”. Please state how these were ensured.

Response 6: This has now been corrected. Researchers avoided questions that could cause any form of psychological trauma.

Comment 7: Demographic characteristics of participants

Page 8, the authors should be consistent with the use of the percentages. Some are in words and others are in figures. Please reconcile.

Response 7: This recommended suggestion has been implemented.

Comment 8: Discussions

• Page 17: instead of using ‘elderly persons in this study’ authors should rather use participants. This should be effected throughout the manuscript.

• Page 19, line 20: The authors should use client instead of patient since not all aged sent to the aged homes for care are sick.

Response 8: This change has been done as recommended.

Comment 9: Strengths of the study

Page 20 line 4: throughout the manuscript, I did not see any recommendation for medical officers to be trained in Gerontology as stated here by the authors. Please delete the medical officers if it has not earlier been stated.

Response 9: This correction has been done.

Comment 10: There are few grammatical errors. I suggest the authors employ the services of a language editor.

Response 10: The help of an editor was sought to make necessary grammatical corrections within manuscript.

Response to reviewer 3:

Comment 1: Title: I suggest that the word “nursing care” in the title be replaced with “healthcare” as findings from the study were not exclusively nursing care issues,

Response 1: This suggestion has been adhered to.

Comment 2: Abstract

The abstract was well structured and well written, but minor revisions required. See main document.

Response 2: Minor correction made in abstract.

Comment 3: Background

The background was well written to set the stage for the study. Very minor revisions required. 

Response 3: Minor correction made in background.

Comment 4: Methods 

This section although it read well, there were few repetitions. Some revisions required. 

Response 4: Repetitions now removed as recommended.

Comment 5: The descriptions of the elderly demographic characteristic did not play any role in the study as it would have been interesting elaborating on that. For example, it was reported that 33% of the older people who were presumed to have been retired were still in active employment. The detail of this was missing. Also, how it impacted their health and medical cost was also left hanging. Furthermore, majority of the older people were females. It would have been interesting to explore and report on the motivation for health seeking behaviour among the two gender and whether marital status and number of children played a role in their healthcare needs and hospital attendance. 

Response 5: This suggestion has been complied with and highlighted within work. However, differences in heath seeking behaviors between females and males was not explored in this study.

Comment 6: The descriptions of the codes (variables) in the themes well not detailed enough. Moreso, most of the themes described were basic healthcare delivery or system challenges and not exclusive nursing care issues. There were some overlaps, for example, the themes financial burden and a cry for cost subsidisation read pretty much the same and could be collapsed as one theme. I believe repetitions and discrepancies will be minimised if further descriptions are given to all the codes/variables. The lengthy quotations should be reduced.

Response 6: The recommendation has been complied with. Overlapping themes have now been condensed to avoid repetitions. There is also now more concentration on health workers rather than just nurses. Lengthy quotations reduced as recommended.

Comment 7: The discussion needs a major revision. The discussion as it is shows a substantial difference of what is discussed and what was reported in the findings. For example, the chunk proportion of information on nursing education in gerontology and licencing examination in that field has no relation to the findings. The study did not look at the courses nurses read in training and therefore has no business discussing that. There is no consistency in the discussion as some of the information given was out of context. 

The study did not look at health service organisation and implications on the care of the elderly, yet, the discussions covers it. I suggest that the discussion section should be rewritten and attention paid to the findings of the study. 

Response 7: The discussion has been rewritten as recommended. Discussions now organised to be in line with findings as recommended in order to stay away from extensive discussions that did not border on findings specifically.

Comment 8: Referencing 

The recommended referencing style and format must be adhered to.

Response 8: Referencing format adhered to as recommended.

---

## [Decision Letter · Decision Letter 2]

10 Jul 2020

PONE-D-19-28368R2

Older people’s challenges and expectations of healthcare in Ghana: A qualitative study

PLOS ONE

Dear MS ATAKRO,  

Thank you for submitting your manuscript to PLOS ONE. After careful consideration, we feel that it has merit but does not fully meet PLOS ONE’s publication criteria as it currently stands. Therefore, we invite you to submit a revised version of the manuscript that addresses the points raised during the review process.

Please make the minor changes to wording requested by Reviewer 2.

We look forward to receiving your revised manuscript.

Kind regards,

Rosemary Frey

Academic Editor

PLOS ONE

Reviewers' comments:

Reviewer's Responses to Questions

**Comments to the Author**

1. If the authors have adequately addressed your comments raised in a previous round of review and you feel that this manuscript is now acceptable for publication, you may indicate that here to bypass the “Comments to the Author” section, enter your conflict of interest statement in the “Confidential to Editor” section, and submit your "Accept" recommendation.

Reviewer #1: All comments have been addressed

Reviewer #2: All comments have been addressed

2. Is the manuscript technically sound, and do the data support the conclusions?

Reviewer #1: Yes

Reviewer #2: Yes

3. Has the statistical analysis been performed appropriately and rigorously? 

Reviewer #1: N/A

Reviewer #2: Yes

4. Have the authors made all data underlying the findings in their manuscript fully available?

Reviewer #1: No

Reviewer #2: (No Response)

5. Is the manuscript presented in an intelligible fashion and written in standard English?

Reviewer #1: Yes

Reviewer #2: Yes

6. Review Comments to the Author

Reviewer #1: (No Response)

Reviewer #2: Title: Fits the content

Abstract

The abstract is well structured and well written.

Background

The background was well written to set the stage for the study.

Methods

This section reads well.

Findings

The findings section was well structured for easy read, however, few observations made and outlined as follows:

1. Under theme 1, the use of verbs such as thoughts, wish, and felt are difficult to measure. Hence, it they must be written as reported expressions. For example, … some participants shared their thoughts that...Others said that they wished….

2. Under theme 2, similar observations were made for example…in line 2 sentence 2 the word felt was used.

3. Under theme 3, the second sentence should begin with the word “Although” instead of though.

4. Under the fourth theme, subtheme 1, the check-ups should read check-up visits from health workers, not by health workers.

Discussion

The discussion reads much better.

Limitation

Well written

7. PLOS authors have the option to publish the peer review history of their article (what does this mean?). If published, this will include your full peer review and any attached files.

Reviewer #1: No

Reviewer #2: **Yes: **LILLIAN AKORFA OHENE

---

## [Author Response · Author response to Decision Letter 2]

10 Jul 2020

Comment 1: Under theme 1, the use of verbs such as thoughts, wish, and felt are difficult to measure. Hence, it they must be written as reported expressions. For example, … some participants shared their thoughts that...Others said that they wished….

Response 1: The recommended change has been complied with and highlighted within manuscript.

Comment 2: Under theme 2, similar observations were made for example…in line 2 sentence 2 the word felt was used.

Response 2: The recommended change has been complied with and highlighted within manuscript.

Comment 3: Under theme 3, the second sentence should begin with the word “Although” instead of though.

Response 3: The recommended change has been complied with and highlighted within manuscript.

Comment 4: Under the fourth theme, subtheme 1, the check-ups should read check-up visits from health workers, not by health workers.

Response 4: The recommended change has been complied with and highlighted.

---

## [Decision Letter · Decision Letter 3]

13 Nov 2020

PONE-D-19-28368R3

Older people’s challenges and expectations of healthcare in Ghana: A qualitative study

PLOS ONE

Dear Dr. Atakro,

Thank you for submitting your manuscript to PLOS ONE. After careful consideration, we feel that it has merit but does not fully meet PLOS ONE’s publication criteria as it currently stands. Therefore, we invite you to submit a revised version of the manuscript that addresses the points raised during the review process.

Please address the issues raised by reviewers three and four.

We look forward to receiving your revised manuscript.

Kind regards,

Rosemary Frey

Academic Editor

PLOS ONE

Reviewers' comments:

Reviewer's Responses to Questions

**Comments to the Author**

1. If the authors have adequately addressed your comments raised in a previous round of review and you feel that this manuscript is now acceptable for publication, you may indicate that here to bypass the “Comments to the Author” section, enter your conflict of interest statement in the “Confidential to Editor” section, and submit your "Accept" recommendation.

Reviewer #2: (No Response)

Reviewer #3: (No Response)

Reviewer #4: All comments have been addressed

2. Is the manuscript technically sound, and do the data support the conclusions?

Reviewer #2: (No Response)

Reviewer #3: No

Reviewer #4: Yes

3. Has the statistical analysis been performed appropriately and rigorously? 

Reviewer #2: (No Response)

Reviewer #3: N/A

Reviewer #4: Yes

4. Have the authors made all data underlying the findings in their manuscript fully available?

Reviewer #2: (No Response)

Reviewer #3: No

Reviewer #4: Yes

5. Is the manuscript presented in an intelligible fashion and written in standard English?

Reviewer #2: (No Response)

Reviewer #3: No

Reviewer #4: Yes

6. Review Comments to the Author

Reviewer #2: (No Response)

Reviewer #3: This study of perceptions of health care among a population of elderly patients addresses an important subject. However, I am concerned that the study lacks a basic hypothesis which leads to a-theoretical purposive sampling and difficulty in interpreting results. It is unclear if saturation was achieved and if so by what criteria; rather, the approach seems to have been to select a qualitative sample size ahead of the study. The paper also does not frame the study in any larger conversations about care for aging populations or quality of care more broadly. Though local by nature, qualitative research is made valuable to various readers through this final process of synthesizing and making connections.

Reviewer #4: Thank you for giving me to review your manuscript. This manuscript is interesting and scientifically meaningful for exploring healthcare challenges and expectations of elderly persons in developing countries facing aging societies in the future. Regarding the contents, I have several suggestions.

1. In the abstract, the authors stated, "Elderly persons across the globe are increasingly becoming marginalised and isolated." This research focuses on developing countries, so the author should focus this sentence on the issues in developing countries.

2. In the introduction, the authors delineate the condition of healthcare among older people broadly. The background section should include developed countries' conditions and the differences from LMICs that the authors deal with in this research for international readers.

3. In the introduction, again, several grammatical mistakes such as the many challenges, very little empirical, and so on. The authors should check wording seriously in the whole manuscript.

4. In the sample section of the method, there is no definition of older persons. There is various definition regarding older people/patients. The authors should show the definition with a reference.

5. In the method section's measurement, the authors clearly described the rigor of the method. To improve the rigor, the authors can include the researchers' reflectivity, such as researchers' backgrounds.

6. This research was based on content analysis, which may not be based on grounded theory. So, a theoretical foundation cannot be achieved based on the authors' methodology. A description of the situation may not be needed.

7. The result section should include a table of the background of the participants.

8. In the limitation section of the discussion, the authors delineate the generalizability of this study. Qualitative research should be assessed regarding credibility, transferability, dependability, and confirmability. This study has a high level of truthfulness. These four criteria should be evaluated through the manuscript.

7. PLOS authors have the option to publish the peer review history of their article (what does this mean?). If published, this will include your full peer review and any attached files.

Reviewer #2: No

Reviewer #3: No

Reviewer #4: No

---

## [Author Response · Author response to Decision Letter 3]

15 Nov 2020

Response to reviewer 3.

Comment 1:This study of perceptions of health care among a population of elderly patients addresses an important subject. However, I am concerned that the study lacks a basic hypothesis which leads to a-theoretical purposive sampling and difficulty in interpreting results.

Response 1: The authors did not formulate a hypothesis in the study. The study used a content analysis approach to explore challenges and expectations of older people in Ghanaian health system and reported this using COREQ criteria for reporting qualitative research (cited in study). The study did not utilise a grounded theory approach and therefore did not use a theoretical sampling technique. These methods were further clarified by the 4th reviewer as she/he states “This research was based on content analysis, which may not be based on grounded theory. So, a theoretical foundation cannot be achieved based on the authors' methodology”.

Comment 2: It is unclear if saturation was achieved and if so by what criteria; rather, the approach seems to have been to select a qualitative sample size ahead of the study.

Response 2: This has now been clarified in the data collection section as follows: Five participants were initially interviewed in each region. Additional five persons were interviewed in each region as saturation was not determined with the initial interviews. Saturation was determined after interviewing the 30th participant.

Comment 3: The paper also does not frame the study in any larger conversations about care for aging populations or quality of care more broadly. Though local by nature, qualitative research is made valuable to various readers through this final process of synthesizing and making connections.

Response 3: The difference in the quality of healthcare received by older people in developed and developing countries has now been stated in abstract to set the tone for the details in differences in background and discussion. The first section of the background is discussed in line with global dynamics in ageing. This has now been given a broader perspective by comparing health systems for the aged in developing countries to developed countries. The difference in care is mostly due to social and economic disadvantages in African countries (this point made and highlighted). The discussion section has also now suggested some evidence-informed global practices that could be adapted in Ghana to improve the health of older Ghanaians. These changes are highlighted in the study.

Response to reviewer 4.

Comment 1: In the abstract, the authors stated, "Elderly persons across the globe are increasingly becoming marginalised and isolated." This research focuses on developing countries, so the author should focus this sentence on the issues in developing countries.

Response 1: This suggestion has been made and highlighted in manuscript (abstract).

Comment 2: In the introduction, the authors delineate the condition of healthcare among older people broadly. The background section should include developed countries' conditions and the differences from LMICs that the authors deal with in this research for international readers.

Response 2: This suggestion has now been included. Aged care systems such as residential and community care in developed countries are now compared with developing and middle-income countries. The changes have been highlighted in the background of the study.

Comment 3: In the introduction, again, several grammatical mistakes such as the many challenges, very little empirical, and so on. The authors should check wording seriously in the whole manuscript.

Response 3: grammatical errors were corrected throughout the manuscript with the assistance of a professional editor.

Comment 4: In the sample section of the method, there is no definition of older persons. There is various definition regarding older people/patients. The authors should show the definition with a reference.

Response 4: The United Nations definition of older persons, that is persons who are 60years or over, has now been included and highlighted.

Comment 5: In the method section's measurement, the authors clearly described the rigor of the method. To improve the rigor, the authors can include the researchers' reflectivity, such as researchers' backgrounds.

Response 5: This suggestion has now been included and highlighted. The background of the authors as qualitative researchers further helped in ensuring qualitative rigor in the study.

Comment 6: The result section should include a table of the background of the participants.

Response 6: A table has now been included and highlighted.

Comment 7: In the limitation section of the discussion, the authors delineate the generalizability of this study. Qualitative research should be assessed regarding credibility, transferability, dependability, and confirmability. This study has a high level of truthfulness. These four criteria should be evaluated through the manuscript.

Response 7: These suggested corrections have now been done and highlighted.

---

## [Decision Letter · Decision Letter 4]

2 Jan 2021

Older people’s challenges and expectations of healthcare in Ghana: A qualitative study

PONE-D-19-28368R4

Dear Ms. ATAKRO,

We’re pleased to inform you that your manuscript has been judged scientifically suitable for publication and will be formally accepted for publication once it meets all outstanding technical requirements.

Kind regards,

Rosemary Frey

Academic Editor

PLOS ONE

Additional Editor Comments (optional):

Reviewers' comments:

Reviewer's Responses to Questions

**Comments to the Author**

1. If the authors have adequately addressed your comments raised in a previous round of review and you feel that this manuscript is now acceptable for publication, you may indicate that here to bypass the “Comments to the Author” section, enter your conflict of interest statement in the “Confidential to Editor” section, and submit your "Accept" recommendation.

Reviewer #2: All comments have been addressed

Reviewer #4: All comments have been addressed

2. Is the manuscript technically sound, and do the data support the conclusions?

Reviewer #2: Yes

Reviewer #4: Yes

3. Has the statistical analysis been performed appropriately and rigorously? 

Reviewer #2: N/A

Reviewer #4: Yes

4. Have the authors made all data underlying the findings in their manuscript fully available?

Reviewer #2: Yes

Reviewer #4: Yes

5. Is the manuscript presented in an intelligible fashion and written in standard English?

Reviewer #2: Yes

Reviewer #4: Yes

6. Review Comments to the Author

Reviewer #2: Title: Fits the content

Abstract

The abstract is well structured and well written.

Background

The background was well written to set the stage for the study.

Methods

This section reads well.

Findings

The findings section was well structured for easy read.

Discussion

The discussion reads much better.

Limitation

Well written

Reviewer #4: The manuscript has been considerably improved. I think that this paper is suited for inclusion in our journal.

7. PLOS authors have the option to publish the peer review history of their article (what does this mean?). If published, this will include your full peer review and any attached files.

Reviewer #2: No

Reviewer #4: **Yes: **Ryuichi Ohta

---

## [Editor Report · Acceptance letter]

7 Jan 2021

PONE-D-19-28368R4 

Older people’s challenges and expectations of healthcare in Ghana: A qualitative study 

Dear Dr. Atakro:

I'm pleased to inform you that your manuscript has been deemed suitable for publication in PLOS ONE. Congratulations! Your manuscript is now with our production department. 

Kind regards, 

on behalf of

Dr. Rosemary Frey 

Academic Editor

PLOS ONE